# U.S. Consumer Demand for Plant-Based Milk Alternative Beverages: Hedonic Metric Augmented Barten’s Synthetic Model

**DOI:** 10.3390/foods10020265

**Published:** 2021-01-28

**Authors:** Tingyi Yang, Senarath Dharmasena

**Affiliations:** 1Department of Economics, Shaanxi Normal University, No. 620, West Chang’an Avenue, Chang’an District, Xi’an 710119, China; 2Department of Agricultural Economics, Texas A&M University, 210T AGLS Building, 2124 TAMU, College Station, TX 77843-2124, USA; sdharmasena@tamu.edu

**Keywords:** plant-based milk alternative beverages, hedonic metric approach, Barten’s synthetic model, consumer demand estimation, Nielsen Homescan data

## Abstract

Consumers in the U.S. increasingly prefer plant-based milk alternative beverages (abbreviated “plant milk”) to conventional milk. This study is motivated by the need to take into consideration varied nutritional and qualitative attributes in plant milk to examine consumers’ purchasing behavior and estimate demand elasticities which are achieved by a new approach combing hedonic pricing model with Barten’s synthetic demand system. The method of estimation is enlightened from the common practice of companies differentiating their products in multidimensions in terms of attributes. A research dataset was uniquely created by associating the products’ purchase data from Nielsen Homescan dataset with exclusive first-hand nutritional data. Estimations began with creating a multidimensional hedonic attribute space based on the qualitative information of different types of plant milk and conventional milk available to consumers and then calculating the hedonic distances by Euclidean distance measurement to reparametrize Barten’s synthetic demand system. Estimation results showed that the highest own-price elasticity pertained to soy milk which was −0.25. Three plant milk types had inelastic demand. Soy milk exerted substituting effects on all types of conventional milk products and vice versa. Soy milk, rice milk and almond milk entertained complementary relationships between each other and four types of conventional milk were strong substitutes within the group.

## 1. Introduction

North America has experienced a huge decline of milk consumption per capita over the past decade, in spite of its uniformly recognized nutritional benefits and sweeping recommendations for consuming milk [1,2]. In the meantime, the market of plant-based milk is expanding with the introduction of vegetative ingredients pervasively known as soy, almond and rice [3]. As argued by Chalupa-Krebzdak et al. (2018) [4], it is the emergence of plant-based milk alternatives that contributes to the slump of bovine milk consumption in North America. Allied Market Research reported that the global market of plant-based milk beverages is expected to cumulate $21.7 billion by 2022, with compound annual growth rate (CAGR) reaching 13.3% from the year 2016 to 2022. Availing of this opportunity, the beverage industry has enjoyed a huge development given the introduction of an extensive array of natural, nutritious, handy and functional beverages in which plant milk is and will likely remain a powerful force promoting the market race. Plant-based beverages are often presented as a healthy, sustainable and animal-welfare-friendly alternative [5,6], and generally can be classified into four categories which are: cereal-based, legumes-based, nut-based and pseudocereals-based [7]. This work mainly focuses on three most common plant milk types: soy milk, almond milk and rice milk.

The main driving forces of the surging demand of nondairy beverages are the inherent deficiencies of traditional dairy products, taking lactose intolerance and protein allergies as examples, and prevalence of vegetarian food [8,9]. The emergence of the flexitarian lifestyle creates great opportunities for the innovation and new product development of plant milk [10]. Among all the influencing factors, changing consumer lifestyle has a far-reaching impact on the functional food and beverage market, but the aging of the population with rising chronic-degenerative diseases and swelling healthcare expenditure also play pivotal parts [11,12]. Undoubtedly, health and nutritional benefits are treated as the most vital factors for consumers of plant-based food products [13]. Plant milks are good sources of macronutrients and micronutrients required by the human body, making it an alternative to conventional milk. For instance, soybean is considered as a complete food which is the raw material used to produce soy milk consisting of protein, carbohydrates, fat and an impressive series of vitamins and minerals. Additionally, soybean is low in saturated fat and contains no cholesterol. These functional beverages are designed to not only satisfy thirst but also supply plentiful vitamins, minerals, proteins and favorable fatty acids [14].

The dominance of soy milk in the market is gradually eclipsed by the emergence of alternative products from other plant sources such as coconut, oat and almond [15]. The rising consumption and continuous diversification of plant milk appeal to an increasing research pursuit of purchasing and consumption patterns of these new products, which is of great importance to manufacturing and marketing the products as well as food policy making. However, little research has investigated consumers’ willingness to pay for consuming the main nutritive components in plant milk. Lacking such information is a critical unmet demand for these milk alternatives bearing the potential to promote health and nutritional status of a population [7,16] as well as food industry development. More importantly, few studies have analyzed consumer demand on the newly developed plant milk based on the differences of nutritional and qualitative attributes in these products. Therefore, this work attempts to shed new light on the major issues of what is the consumption pattern of plant milk and whether there is a significant substitution effect between plant milk and conventional milk. This relies on evaluating the nutritional deviations of the attributes present in plant milk and conventional milk and estimating consumer demand using cutting-edge modifications to demand system models based on hedonic metric approach developed by taking into consideration characteristic differences. Income and price elasticities were estimated using data recording consumers’ purchasing activity during the period 2004–2015. The main objectives of this research are: (1) create an attribute-space hedonic matrix for each product category associated with a preidentified group of brands at Universal Product Code (UPC) and brand level; (2) estimate consumer demand for plant milk categories covering almond milk, soy milk, rice milk and dairy milk constituting whole milk, fat-free milk, 2% fat milk and 1% fat milk using attribute space hedonic metric augmented Barten’s synthetic model, and (3) examine price sensitivities and substitute and complement effects among the aforementioned beverages.

We briefly explain the rationale of specifying the hedonic augmented Barten’s synthetic model for later use in modeling the demand of plant-based alternative beverages. The theory of consumer demand is about individual behavior with respect to the choice of quantities of a potentially large number of elementary goods [17]. It is assumed that a representative consumer faced with different alternatives chooses a certain product to maximize utility. From an economic standpoint, consumer demand cannot be analyzed without referring to the price of the commodity and the corresponding quantity with their relationship defined by a certain demand equation. Most earlier demand analyses such as the works by Schultz (1938) [18] are partial demand examinations where food price and income per capita are regarded as decisive variables in a demand equation, but they did not chew over the complete intermutual nature of food demand [19]. Also, as Thomas (1987) [20] mentioned, the single-equation approach to model consumer demand is insufficient due to the absence of a sound theoretical foundation and incapability to generate consistent parameter estimates such as cross-price elasticities [20]. Therefore, food demand research is requested to implement a complete demand system approach in order to explicitly recognize the communal demand relationships among all foods.

The development of a demand system is started from specifying and estimating the following system: *q* = *q*(*m*,*p*), where *q* is an *n*-vector of quantities of *n* commodities and services and *m* = *p’q*, where *m* is the total expenditure on the *n* goods considered. In other words, consumers’ preference is represented by the cost function, *c*(*u*,*p*), which specifies the minimal expenditure imperative to fulfilling a specific utility, *u*, at given prices, *p* [21]. Therefore, based on the utility function which includes the employment of indirect utility and application cost functions, the ordinary demand system can be derived. The concept of a hedonic pricing model also originated from the reconsideration of utility measurement. Lancaster (1966) [22] contended that it is product attributes/characteristics which the good occupies that engender utility and not just the quantity of the good consumed. Therefore, using the parameters estimated from hedonic pricing models to reparametrize a demand system is theoretically justifiable.

Differently from previous research, the demand system we applied in this study is rooted in the particular function form of Barten’s synthetic demand model [23]. Specifically, hedonic pricing models were estimated for seven products (soy milk, almond milk, rice milk, fat-free milk, 1% milk, 2% milk and whole milk) considered in the first place and then the estimated coefficients were used to obtain the marginal value (shadow price) of each qualitative attribute through a simple multiplication procedure. Barten’s synthetic demand system was reparametrized by the factors derived from a multidimensional hedonic attribute space built on the marginal value of attributes based on which the hedonic distances were calculated applying Euclidean distance measurement.

The merits of this work can be embodied by the following aspects: In the first place, it contributes to manifesting the expenditure patterns of plant milk based on the estimated own-price, cross-price and expenditure elasticities of three beverage categories which hopefully would contribute to forecasting the future advancement of consumption in terms of these beverages and providing useful references for manufacturers of plant milk to explore and expand the corresponding market, develop differentiated products and formulate a marketing strategy. The estimation results are also crucial for advertisers to strategically position plant milk in the competitive conventional milk marketplace.

Even though the inclination toward plant milk is ubiquitously observed, few works have been done to reveal how sensitive consumers are to price changes, which is of great importance for companies to adjust their pricing strategies and production plan accordingly, simultaneously taking into consideration competitor’s behavior. Currently, the global food market has already detected the business opportunities related to plant milk and put their tentacles in this industry, predictably leading to fiercer competition in the near future. Therefore, in order to barge itself to the forefront, a company should be fully informed with consumers’ attitudes and purchasing behavior and then keep wits about the fast-changing market where demand elasticities play a crucial part in exhibiting the consumption structure, which is affected by the income and price changes of a certain group of products and also the eating habit and consumption concept. Demand and income elasticities of plant milk estimated in this research would provide strong evidence showing a renewal of consumption concept of U.S. consumers favoring a healthy and nutritious diet which is characterized by less fat and sugar intake, high protein and nutrition balance. This transformation is about to drive more companies to devote themselves into this emerging market and encourage new product development to consistently improve health conditions of people, creating a virtuous circle not only for human development but also for social advancement.

Additionally, to the best of our knowledge, no research has been done to incorporate the information of consumers’ willingness to pay for qualitative attributes into demand systems and systematically estimate the demand elasticities of the three most common types of plant milk. The augmented Barten’s synthetic demand model we developed achieved a great improvement of the traditional demand model such that it addressed the price endogeneity issue common with estimating a complete demand system and, more importantly, scaled down the number of parameters required to be estimated. In light of such advantages, the model can be extensively applied to analyze demand of other qualitatively differentiated products.

## 2. Literature Review

Research about plant milk has proliferated in recent years with the product increasingly becoming prevalent in the market. Some review articles about plant-based alternative beverages, for instance, Vanga and Raghavan (2018) [24] delineated the nutritional distinctions among different plant milk and conventional milk products and by comparison argued that nutritional soy milk is the best substitute to replace dairy milk in consumers’ diet. Singhal et al. (2017) [25] reviewed and compared the contents and nutritional value of nondairy beverages to cow’s milk. Sethi et al. (2016) [26] outlined the functional constituents of plant milk, their health benefits and the technological interventions ought to be implemented to enhance the quality and trustworthiness of plant milk. Fuentes and Fuentes (2017) [27], relying on the concepts of the marketing device and qualification, described, conceptualized and critically discussed the ways and consequences of constructing a mass market for vegan milk alternatives. Mäkinen et al. (2016) [12] gave an overview on the technological, nutritional and environmental aspects of plant milk substitute production and consumption.

In order to capture consumers’ attitude, preferences and motivation toward plant milk and their related influencing factors, some empirical research has sprout up to achieve this goal. Empirical consumer research by Laassal and Kallas (2019) [28] utilized Homescan data of 343 households and drew on revealed preference discrete choice experiment to evaluate consumers’ propensity to dairy alternative products in Catalonia. The main results implied that price was the principal contributing factor, followed by flavor attribute. By conducting a quantitative survey of plant milk and cow’s milk consumption in Austria, Haas (2019) [29] analyzed consumers’ perception of products’ image and contrasted the motivational configuration behind consumption behavior. The result showed that consumers of plant milk gave more credits to plant milk, ascribing to its better digestibility and the allergy-free merit, with consumption motivations more diversified to have animal welfare and environmental sustainability involved. Dharmasena and Capps (2014) [30] explored consumer demand for soy milk, white milk and flavored milk.

In common practice, consumer demand of agricultural food products is analyzed with respect to estimating price elasticities and expenditure elasticities. Many empirical articles in the extant literature apply conventional demand models, such as the almost ideal demand system (AIDS) [31] and the Rotterdam model [32] in which own-price, cross-price and expenditure elasticities are estimated based the exploitation of the relationship between prices and market shares. Traditional demand models assume that consumers’ utility is gained through the quantity of a specific good they consume, without considering the intrinsic properties of a particular good which distinguishes it from other similar goods. While, based on the characteristics approach to estimating demand, the random utility model (RUM) has been widely used as an alternative model to the conventional demand model to estimate consumer demand, it is very complicated computationally and difficult to estimate [33,34,35,36]. Given this issue, Pinske et al. (2002) [37] developed a distance metric (DM) approach which uses spatial distance to the desired characteristic to estimate price elasticities for different products. Compared to the RUM approach, which requires a simulation process, the DM approach is more straightforward to employ and moldable enough to feature the substitution patterns within differentiated products [34]. Rojas and Peterson (2008) [38] applied this technique to the retail beer market by selecting alcohol content as the primary quality criterion of beer products supplemented by other different distance combinations. However, one of the prominent weaknesses of their work is that the distance measurement constructed is obscure and is built on prior knowledge of the data, specifically in selecting the base category of the product in developing the distance matrix.

Some academic studies were generated in order to remedy the research limitations aforementioned, specifically, the selection of the arbitrary base category in the construction of the distance matrix. For instance, Gulseven and Wohlgenant (2015) [34] proposed a hedonic metric (HM) approach to examine and estimate the demand for retail milk, showing that this approach alleviates the embedded ambiguity in selecting the base category problem in the DM model and considerably curtails the number of parameters necessary to estimate a demand system, achieving the possibility of incorporating abundant discriminated products in each demand system. The study of Gould and Lin (1994) [39] borrowed Lancaster’s (1966) [22] idea of a hedonic framework, but its motivation is to estimate demand for qualitative attributes dependent on products’ elasticity, whereas this work aims to estimate consumer demand based on their qualitative characteristics. Even though Gulseven and Wohlgenant (2015) [34] have developed a method to estimate own-price, cross-price and expenditure elasticities of several conventional and soy milk products based on quality attributes and hedonic prices, the milk product types included in their study are limited and mingled with dairy alternative beverages, since soy milk is not commonly categorized into conventional milk categories. Also, their method has many ambiguities in practice. For example, they do not explain clearly how the hedonic matrix is constructed and the own closeness index has application errors.

In view of the shortcomings of the above research, the remaining sections of this study are organized as follows: Based on the hedonic pricing model estimation of each out of seven products at brand level, a hedonic attribute space was constructed. After obtaining all the parameters determined by the hedonic distance matrix which captures the Euclidean distance between two products, the hedonic metric augmented Barten’s synthetic demand system was estimated to derive expenditure, own-price and cross-price elasticities to uncover the price sensitivities and substitute and complement effects among three plant milk and four types of milk products, expectedly providing information for manufacturers, retailers, advertising companies and nutritionists for strategic decision making.

## 3. Model Specification

We start by discussing the hedonic pricing model which was estimated in priority to obtain the marginal value (or shadow price) of each quality attribute attainable from the product. Then, the shadow price was employed as a weight added onto the amount of each characteristic to measure the value added from each attribute. After this, a hedonic attribute space was created where each component in the space was the hedonic attribute distances between any two attributes considered in this study. The name hedonic attribute space was created by analogy to Euclidean space, where multiple attributes in products constitute a multidimensional space and each attribute distance between two products can be treated as a point in the space. Following this measurement, the hedonic distance of two products finally composed the hedonic distance matrix based on pairwise comparison of Euclidean distance between two products. After each differentiated product was allocated into the hedonic attribute space and then the hedonic distance matrix, we could reparametrize the price coefficients in Barten’s synthetic demand system.

Hedonic regressions are being increasingly used to better understand the drivers of prices for consumer products (Sopranzetti, 2015) [40]. One purpose of the hedonic method might be to obtain estimates of the willingness to pay for or marginal cost of producing the different characteristics [41]. Here we focus on the first main purpose to acquire consumers’ willingness to pay for the different attributes of plant milk. X = ($X_{1},X_{2}\ldots ,X_{l})$ represents the qualitative characteristic combination. The functional dependency of the price of a product on its characteristics vector *X* can be generally modeled by $P=f\left(x\right)+\varepsilon$, where ε is the error vector and P is the observed price.
(1)$$P_{i}=\beta_{0}+\underset{m}{\sum }\beta_{m}A_{im}+\underset{n}{\sum }D_{n}X_{in}+\varepsilon_{i},\;\;i=1,\;2,\ldots ,7$$
(2)$$\ln(P_{i})=\beta_{0}+\sum_{m}\beta_{m}A_{im}+\sum_{n}D_{n}X_{in}+\varepsilon_{i},\;\;i=1,\;2,\ldots ,7$$
where $A_{im}$ is the amount of nutritional attribute m, including calories, fat, calcium, vitamin A, vitamin D, fiber and protein contained in product i, thus m = 7 and i = 7. $X_{in}$ is other factor(s) that might affect prices which are brands, coupon, deal, package size, multipack and yearly dummies (where l=1, 2,…, 42 and m+n=49). The linear and semi-log hedonic pricing models are the two most commonly applied forms. The hedonic pricing model has been described by Palmquist and Smith (2002) [42] as “one of the ‘success’ stories of modern applied microeconomic analysis.” When variables are omitted or replaced by proxies, simpler forms such as the linear, semi-log, double-log and the Box-Cox linear perform the best [43]. The semi-log structural hedonic pricing model is far superior to its linear counterpart for permitting the value of a given attribute to vary proportionately with the value of others (Sopranzetti, 2015) [40]. In order to make no assumptions exerted on the model form, the models we incorporate to fit the data include both linear and semi-log forms.

Linear hedonic pricing model assumes that the relationship between prices and attributes is linear. As shown in Equation (1), price of a good i is a function of the sum of attribute values [44]. As such, the total value of each attribute is derived by multiplying quantity of the attribute by shadow price of that attribute. $P_{i}$ is the monthly average price in the time period. If a semi-log functional form for price–attribute relationship is presumed [45], as indicated by Equation (2), then the model is interpreted as the dependence of log-price of the product on related characteristics. Again, all the attributes can be separated into nutritional attributes $A_{im}$ and other related attributes $X_{in}$. Accordingly, $\beta_{m}\;$and$\;D_{n}$ are the coefficients (implicit prices of attributes) to be estimated. $\beta_{0}$ denotes the intercept and $\varepsilon_{i}$ represents the stochastic error term. Because the data applied in this study were pooled consumer data, where time factors might play a role in the price changes, we took time effects into consideration by adding yearly dummies into the model. Other influencing variables embraced included values of the multipackage, package size, coupons, deals, brands, etc.

### 3.1. Hedonic Attribute Space and Hedonic Distance Measurement

The practice of putting products in a certain space and measuring their difference based on their locations in the space can be traced back to the famous model, namely, the Hotelling Model, which was developed by Hotelling (1929) [46], who aimed to explain the Bertrand Paradox in oligopolistic competition. He suggested that the difference of two products can be considered as the distance between two companies that produce the products. This is the first argument taking into consideration the distance to measure the difference between two products. Therefore, putting products into a space and measuring their difference based on distances—the main idea in hedonic distance measurement—is theoretically justifiable. This idea is supported by Pinske, Slade and Brett (2002) [37], who use a vector $d_{ij}\;$to measure the distance between regions or outlets i and j in some metric. The geographic locations of the two outlets is measured by the Euclidean distance and a zero/one variable that indicates whether j is i’s nearest neighbor. It is suggested that more relevant applications include the assessment of proximity in taste space. Employing this method, Rojas and Peterson (2008) [38] constructed a distance metric where cross-price and cross-advertising coefficients ($b_{jk}$ and $c_{jk}$) in the adjusted AIDS model are defined as functions of different distance measurements between brands j and k. They suggest that these distance measurements may be either continuous or discrete. Alcohol content of a brand, for example, can be applied to establish a continuous distance measure which can be measured by an inverse measure (reciprocal in math) of Euclidean distance, or closeness, in product between brand j and k.

However, a big issue appearing during the application of the distance metric method is that the continuous distances’ computation is dependent on preimposed attributes and selecting the base category, which, most of the time, is arbitrarily decided by the researcher, leading to inconclusive estimation results. Therefore, in order to remedy the limitations embedded in the distance metric method, we applied the distance measurement to construct a hedonic attribute space based on the coefficients estimated in hedonic pricing models. The detailed procedure to acquire a hedonic attribute space can be referred to in Appendix A. The final results of the hedonic distance matrix constructed are presented in Table 1. The rescaled continuous hedonic distance valued between 0 and 1 is shown in Table 2.

Referring to the concept of closeness and closest neighbors in the above distance metric method, the nearest neighbor of two products is defined by two products located next to each other in the hedonic space. Following this proposition, the nearest neighbor (closest product) for each product can then be directly acquired from Table 3, where the shortest distance in a linear hedonic matrix between almond milk and other products is associated with soy milk (distance value 0.26). With the same logic, for soy milk, rice milk, 2% milk, 1% milk, whole milk and fat-free milk, the closest products were fat-free milk, fat-free milk, 1% milk, soy milk, soy milk and rice milk, respectively. Table 4 is drawn to present these results in a more straightforward way in which for each product, its nearest neighbor was valued at 1 and that was how the value of cross-product closeness index $d_{ij}^{nn}$ was assigned. In a word, if two products, i and j, are nearest neighbors, the cross-product closeness index is 1, otherwise, it is 0.

### 3.2. Own-Closeness Index Measurement

Borrowing the idea of Sabidussi (1966) [47] and Wang and Tang (2014) [48] to construct the closeness index, we first built a network which contained notes and directed edges. According to the social network paradigm, for each edge denoted as $\left\{i,j\right\}\in E$, there is a corresponding edge $\left\{j,i\right\}$. In this sense, the seven products considered in this study could be treated as seven nodes, and each node was connected with the other six nodes to have 42 edges in total. For convenience, let $V_{\left(i,j\right)}$ and $V_{\left(j,i\right)}$ signify, respectively, the sets of nodes that were connected. V can be separated into $\left\{i\right\}\;\mathrm{and}\;V_{\left(ij\right)},\;\forall j\in N_{i}$, where $N_{i}=\left\{j\in V:\left\{i,j\right\}\in E\right\}$ denotes the set of neighbors of node i. Then, the classic closeness denoted as$\;C_{i}$ of the node i∈V can be represented as:(3)$$C_{i}≜\frac{N-1}{\sum_{j\in V}d_{ij}}$$
where $d_{ij}=d_{ji}$ is the distance (i.e., length of the shortest paths) between two nodes i and *j*, and the factor N−1 in the numerator is inserted to make $C_{i}\in \left(0,1\right)$. It is suggested that the larger the $C_{i}$, the closer node i is, on average, to all other nodes in the graph G. Closeness centrality, originally defined by Sabidussi (1966) [47], is a basic centrality measure which designates how centrally located a node is. It attaches higher scores to nodes which have shorter distances to all other nodes. Relating closeness centrality to the classic closeness expression (Equation (3)), the larger $C_{i}$, the more central the node is, and this means the corresponding product has comprehensively the smallest distances with all the other products. As Figure 1 presents, each product is a node and connected with other nodes by a symmetric edge with the length of the edges being the Euclidean distance. A more intuitive way to illustrate the closeness centrality is shown on the right of Figure 1, in which the larger $C_{i}$ of the product is, the bigger node i is drawn.

### 3.3. Reparameterization in Demand System

Demand system models are widely used in estimating demand relationships in a wide range of food products. The almost ideal demand system (AIDS) and Rotterdam model are two commonly used such demand systems. One of the major advancements in demand system modeling was the development of the Rotterdam model by Theil (1965) [32] and Barten (1964) [49]. Barten (1993) [23] proposed Barten’s synthetic model (BSM), incorporating the differential versions of the almost ideal demand system (AIDS) model [31], the National Bureau of Research (NBR) model [50], the Rotterdam model [32,49] and the Dutch Central Bureau of Statistics (CBS) model [51]. Following Matsuda (2005) [52], the basic BSM is as follows:(4)$$w_{i}d\ln q_{i}=\left(\beta_{i}+\lambda w_{i}\right)d\ln Q+\sum_{j=1}^{n}[\gamma_{ij}-\mu w_{i}\left(\delta_{ij}-w_{j}\right)]d\ln p_{j}$$
where *i* = 1, 2, …, *n* and $\beta_{i}\;\equiv \;\left(1-\lambda \right)b_{i}\;+\lambda c_{i}$ and $\gamma_{ij}\;\equiv \;\left(1-\mu \right)s_{ij}+\mu r_{ij}$, $w_{i}\equiv \frac{p_{i}q_{i}}{m}$ denotes the expenditure share of good i, which determines the allocation of additional expenditure to the good, $d\ln Q\equiv \sum_{i}w_{i}d\ln p_{i}$ denotes the Divisa volume index.$\;\delta_{ij}\;$is the Kronecker delta, which is equal to unity if i=j and zero otherwise; $b_{i}$ are constant coefficients. After imposing the restrictions on coefficients μ and λ, we retrieve the LA/AIDS, the Rotterdam, the CBS and the NBR models. Specifically, (λ, μ) = (0, 0) gives rise to the Rotterdam model; (λ, μ) = (1, 0) creates the CBS model; (λ, μ) = (0, 1) yields the NBR model; (λ, μ) = (1, 1) generates the AIDS model. The reparameterization of a demand system and then derivation of own-price elasticities, cross-price elasticities and expenditure elasticities can be found in Appendix B.

## 4. Data

Nielsen Homescan consumer panel data 2004–2015 was exploited in this research. The analysis of this research is partly based on data from The Nielsen Company (U.S.), LLC and marketing database offered by the Kilts Center for Marketing Data Center at the University of Chicago Booth School of Business. The conclusions extracted from applying the Nielsen data are those of the researchers and do not reflect the views of the Nielsen. Nielsen is not responsible for, had no role in, and is not involved in deriving and arranging the results reported herein. Consumer level data is gathered by tracking households’ purchase behavior. From this large panel data, consumers’ weekly purchase information is extracted. From this database, price, quantity and expenditure data of plant milk and conventional milk were acquired. The final dataset was aggregated to Universal Product Code (UPC) level in order to capture enough variations of nutritional variables. The format of original data file in which individual household’s purchase information was recorded by specific trip dates, so it can be commonly observed in the dataset that purchases may appear many times each month or even no purchase activities at all. Additionally, the products’ nutritional attributes and their proportions are barely changed from time to time, so constructing time series data cannot ensure enough variations of nutritional attributes for hedonic pricing model estimation. Therefore, the best choice available was to transform the dataset into pooled data form at UPC level. The structure of the final dataset was characterized by product types (soy milk, almond milk, rice milk, 1% milk, 2% milk, fat-free milk and whole milk) which comprise different products recorded by the UPC code.

### 4.1. Data Manipulation for Estimating Hedonic Pricing Models

It is one of the trickiest issues to gather data about nutritional information of plant milk for hedonic pricing model estimation primarily due to the unavailability of an applicable database pertaining to such information. The availability of nutritional information indispensable in this work was directly obtained from the products’ label (i.e., individuals’ visual observation of beverage packages). The final dataset of nutritional attributes reflects the same set of qualitative information that consumers have about these products based on their labels. Other than the nutritional variables including calories, protein, fat, vitamin A and vitamin D, which consumers are concerned with the most when making purchasing decisions, we attempted to find out other variables that might exert significant effects on the products’ prices. For example, apparently, the availability of deals and coupon contributes to lower purchasing prices due to the fact that if a coupon is applied, the price is discounted accordingly. So, a dummy variable which implies if consumers receive a deal or a coupon as incorporated into the model. Additionally, we controlled the time effects by adding yearly dummies considering the fact that the pooled dataset was transformed from time series of consumer purchase data. Brand was also proven to be an important influencing factor to price, so the corresponding dummy variable was added, taking value 1 as a store brand and 0 as a national brand. The final variables of qualitative attributes involved in hedonic pricing models encompassed package size, the multipackage (units per package), deals, coupon, brands and yearly dummies.

Every product appearing in the market was given a unique barcode called the Universal Product Code (UPC), which serves as product identification in the Nielsen Homescan data. Therefore, the nutritional information was collected according to different UPCs which were used to record different products in the Nielsen dataset. Then, the detailed products’ attribute data was merged with Nielsen based on UPCs to formulate the dataset ready for estimating hedonic pricing models. The detailed data manipulation procedure is briefly summarized in Figure 2.

The process to derive monthly average unit price paid (unit price paid in short in Figure 1), dependent variable in hedonic pricing model, is shown in Figure 2. First, we obtained each product’s information such as package size, multipackage, size unit from the file called “products” and then merged the information with the trips file from the Nielsen Homescan dataset to acquire the dataset including key variables: quantities sold, total price paid by consumers, coupon value, deal, multipackage, package size and size unit. Trips file records consumers purchasing information, including detailed information of products purchased and total expenditure in each trip and the trip date. The unit price paid, shown in Figure 2, was calculated by first dividing the variable final price paid by the quantity, where final price paid was calculated by subtracting the value of variables “coupon value” from the value of “total price paid”. Then, the variable monthly average price/oz was gained by taking average of unit prices paid each month and year.

### 4.2. Data Manipulation for Estimating Demand System

As indicated in Figure 2, the process to construct the final dataset to estimate the demand system began with the original datasets which were used to estimate the hedonic regressions. Parameters such as price, quantity and budget share of each product needed to be known before estimating the demand system. First, as Figure 2 shows, monthly average prices and monthly average quantity could be calculated from the products file directly. But this time, the variables price and quantity were averaged by product type level instead of UPC level for demand system estimation. Then, we merged the seven separate data files created previously for estimating hedonic pricing model in order to acquire the total expenditure of all seven product types, which was created by first summing up final expenditure in each trip, which, as shown in Figure 2, was calculated by subtracting “coupon value” from “total price paid” in each trip, and then sum the data again by each product type. The calculation of average budget share $w_{i}$ of each product type each month and each year was dependent on total expenditure of seven product types divided by total expenditure per product type.

From the above steps, we had all the necessary variables at hand to estimate the demand system. Because we had seven product types with average budget shares from year 2004 to 2015, the final dataset for estimating the demand system contained 144 (12 months times 12 years) observations for each product type and the whole demand system had 1008 (144 times 7) observations in total.

## 5. Empirical Results and Discussion

### 5.1. Estimation Results of Hedonic Pricing Models

As mentioned before, the first step of estimating hedonic metric augmented Barten’s synthetic model was to estimate hedonic pricing models. The estimation results of hedonic pricing models generally conformed to our expectations. Both model forms fit well for plant milk and conventional milk data. The detailed estimation results can be found in Yang and Dharmasena (2020) [53] As intuition suggests, consumers’ willingness to pay for a specific attribute will be negative if they are discouraged to accept this given attribute. In the linear hedonic pricing model, the estimated coefficient captures the contribution of a unit increase in the attribute to the change of a unit price change on average. The estimated coefficient of fat content negatively contributed to unit price of rice milk and soy milk, taken the value −0.1080 and −0.0025, respectively, revealing that a unit increase in fat content yielded 10.8 percent and 0.25 percent decrease of consumers’ willingness to pay for a unit price of rice milk and soy milk. Gulseven and Wohlgenant (2015) [34] also analyzed the effect of energy attribute on prices, taking carbohydrates, instead of calories, into consideration. Estimation results showed that carbohydrates have positive influence on prices. Similarly, calories exerted about 0.03 percent positive effect on unit prices of soy milk. Americans are encouraged to take calories as their main source to acquire nutrition and to make wise nutrient-dense choices from all food groups [54]. As concluded by Taubes (2007) [55], good and bad calories coexist, so what matters to good health is not how many, but the kind of calories we take in. Studies have shown that calorie consumption is closely correlated to income, making its elasticity not constant [56,57,58].

As expected, vitamin A positively impacted the prices of soy milk and almond milk, with estimated coefficients valued at 0.0029 and 0.073, respectively. The significant positive effects of protein and calcium on prices unfolded consumers’ preference on these nutritional attributes. The results also show that protein entertained the highest weight, indicating that protein was the most favorable qualitative attribute of soy milk and almond milk, and calories were reluctantly accepted by consumers. These results are consistent with that of Gulseven and Wohlgenant (2015) [34], which show that the coefficient of protein enjoys the highest value, followed by carbohydrates and fat. Rising expenditures or incomes, as demonstrated by Widarjono (2012) [59] and Faharuddin et al. (2014) [60], leads to increased consumption of fats and proteins in comparison with the consumption of calories and carbohydrates.

### 5.2. Estimation Results from Barten’s Synthetic Demand System

As mentioned before, the estimated coefficients from linear and semi-log hedonic pricing models are used to calculate the value-added terms and pair-wised difference in-between characteristics to obtain the hedonic distance matrix. After all the parameters are gathered, we reparametrized Barten’s synthetic model to create the hedonic metric augmented Barten’s synthetic model from which expenditure elasticities, own-price and cross-price elasticities (both uncompensated and compensated) were estimated for the seven products over the 144-month period. We dropped one equation for estimation purposes, as Barten (1969) [61] proposes that parameter estimates are invariant to the dropped equation and the dropped parameters can be recovered from the adding-up restrictions.

Presence of possible autocorrelation (serial correlation) was examined through the autocorrelation and partial autocorrelation function generated for each series. Seasonal (quarterly) dummy variables were significant at 0.01 level for almond milk, soy milk and fat-free milk, as shown in Appendix C, supporting quarterly seasonality present in the dataset. Due to this fact, the following version, shown as Equation (5), of Barten’s synthetic model, in which the disturbance term and quarterly seasonal dummies have been incorporated to represent seasonal adjustment.
(5)$$\begin{array}{r} w_{it}d\ln q_{it}=\;\left(\beta_{i}+\lambda w_{it}\right)d\ln Q_{t}+\alpha_{0}d\ln p_{it}+\left(\alpha_{1}x_{i}^{s}+\alpha_{2}x_{i}^{c}-\mu w_{i}\left(\delta_{ii}-w_{it}\right)\right)d\ln p_{it}\\ +\sum_{i\neq j}[c_{h}d_{ij}^{h}+c_{nn}d_{ij}^{nn}\;–\left.\mu w_{it}\left(\delta_{ij}-w_{jt}\right)\right]d\ln p_{jt}+\sum_{j=1}^{3}d_{j}Q_{ijt}+e_{it}\; \end{array}$$
where *i* = 1, 2, …, 7 indexes seven products in the demand system; *t* indexes the time in months over 12 years, i.e., *t* = 1, 2, 3, …, 144; $p_{jt}$ are monthly average prices for each milk product; $q_{it}$ is quantity (oz.) consumed of each milk product; $Q_{ijt}$ is the quarterly dummy used to capture the seasonality relating to four quarters in all the years. Monthly budget share of each plant-based milk alternative beverage consumed is denoted by $w_{it}$ where $w_{it}=\frac{p_{it}q_{it}}{m}$. Additive disturbance term is denoted by $e_{it}$.

Calculated autocorrelation and partial autocorrelation functions of the residuals pertaining to all plant milk confirms the presence of serial correlation. The result conformed to expectation due to the time-series nature of the dataset. A close study of the data indicated the presence of a fifth-order autoregressive process of disturbance terms in the system. Each demand equation was fitted with a first-, second-, third-, fourth- and fifth-order autoregressive process of disturbance terms and simultaneously, the significance of each autocorrelation coefficient was investigated. After all the above exercises were finished, it was proved that disturbance terms behave as an *AR (5)* process. Accordingly, Barten’s synthetic model was fitted by the following specification assuming the existence of a disturbance process:(6)$$e_{it}=\rho_{i1}e_{i,t-1}+\rho_{i2}e_{i,t-2}+\rho_{i3}e_{i,t-3}+\rho_{i4}e_{i,t-4}+\rho_{i5}e_{i,t-5}+u_{it},\;$$
where $\rho_{i1}$, $\rho_{i2}$,$\;\rho_{i3}$, $\rho_{i4}$ and $\rho_{i5}$ represent first-, second-, third- and fourth-order autoregressive parameters, respectively.$\;u_{it}$ is the white-noise disturbance (independently and identically distributed with zero mean and constant variance). Lastly, the form of the reparameterized Barten’s synthetic model taking *AR (5)* disturbances into consideration is shown as:(7)$$\begin{array}{c} w_{it}d\ln q_{it}=\overset{5}{\underset{k=1}{\sum }}\rho_{j}{\left(w_{it}d\ln q_{it}\right)}_{t-j}+(\beta_{i}+\lambda w_{it})d\ln Q_{t}+\alpha_{0}d\ln p_{it}+(\alpha_{1}x_{i}^{s}+\alpha_{2}x_{i}^{c}-\mu w_{i}(\delta_{ii}-w_{it}))d\ln p_{it}+\\ \underset{i\neq j}{\sum }[c_{h}d_{ij}^{h}+c_{nn}d_{ij}^{nn}\;-\mu w_{it}(\delta_{ij}-w_{jt})]d\ln p_{jt}-\overset{5}{\underset{k=1}{\sum }}\rho_{k}\{(\beta_{i}+\lambda w_{it-1})d\ln Q_{t-k}+\alpha_{0}d\ln p_{it}+\\ (\alpha_{1}x_{i}^{s}+\alpha_{2}x_{i}^{c}-\mu w_{it-k}(\delta_{ii}-w_{it-k}))d\ln p_{it-k}+\underset{i\neq j}{\sum }[c_{h}d_{ij}^{h}+c_{nn}d_{ij}^{nn}]\;–[\mu w_{it-k}(\delta_{ij}-w_{jt-j})]d\ln p_{jt-k}\}+\\ \overset{3}{\underset{j=1}{\sum }}d_{j}Q_{ijt}+e_{it} \end{array}$$
Correlation and covariance matrix of log prices of seven products is shown in Appendix C. We estimated all models with no restrictions imposed from demand theory, but tested for symmetry and homogeneity later. Linear and semi-log model estimations in the demand system are shown in Appendix C. In the linear case, there were 22 out of 37 parameters estimated showing significance at *p*-value 0.05. The parameter estimates $a_{0}$, $a_{2}$,$\;c^{h}$, $c^{nn}$ and $b^{1}$ were significant. Calculated autocorrelation coefficients were statistically significant at 99% level, indicating the presence of *AR (5)* disturbance terms. The joint hypotheses test for seasonal dummies, λ (lambda) and μ (mu) are shown in Table 4. The test of homogeneity failed to reject six out of seven homogeneity restrictions, null hypothesis being homogenous of degree zero in price and expenditure. However, the symmetry test demonstrated mixed results. Moreover, the joint hypotheses for λ (lambda) and μ (mu) were rejected for possibility of differential demand systems with Rotterdam, AIDS, NBR and CBS versions. Hence, it could be concluded that Barten’s synthetic model itself is an appropriate demand model to model this data (see also Matsuda, 2005). The parameter estimates and joint hypotheses test of demand system using semi-log hedonic metric approach showed similar results.

Three out of seven budget share series were nonstationary, indicating the sample mean over 144 observations were both candidates of local coordinates to evaluate elasticities. In view of this, we tried to obtain expenditure elasticities using the last 12 observations of each budget share shown in Appendix D. Table 5 shows the expenditure elasticities and uncompensated own-price and cross-price elasticities calculated using the last 12 observations of budget share, respectively. The estimates of compensated own-price and cross-price demand elasticities are shown in Table 6. It is shown that the calculated expenditure elasticity estimates, except for rice milk, were significant at or above the *p*-value 0.05. Soy milk was found to be the most expenditure-elastic plant-based milk alternative beverage. Because expenditure elasticity performs as a measure of the responsiveness of expenditure on, or consumption of, a product to a change in real income, this result also indicates that soy milk was the most responsive plant-based milk alternative beverage for varying total expenditure values. Almond milk also had high expenditure elasticity (3.60). This is consistent with the results of Paraje et al. (2016) [62], Guerrero-Lopez et al. (2017) [63] and Chacon et al. (2018) [64], which shows that demand for beverages was responsive to total expenditure changes. Expenditure elasticities with respect to soy milk and almond milk were high due to the fact that their expenditure shares were low (expenditure shares are in the denominator of the expenditure elasticity calculation and this could explain why the expenditure elasticity for almond milk and soy milk were somewhat high). In terms of conventional milk products, they were all expenditure inelastic (2% milk 0.83; fat-free milk 0.57; whole milk 0.55) except for 1% milk (1.14), indicating that they are normal goods.

All uncompensated and compensated own-price elasticity estimates presented negative signs, which successfully indicated theoretically consistent own-price elasticity estimates. The estimates were statistically significant except a few. Compensated own-price elasticity of demand for soy milk was −0.25, which was the highest, indicating that consumers are highly insensitive to own-price changes. Gulseven and Wohlgenant (2015) [34] also found that soy milk entertained the highest own-price elasticity, followed by 1% milk, full-fat milk, skim milk and 2% milk. Higher own-price elasticity of demand for soy milk is attributed to small budget share and high prices associated with soy milk. In other words, marginal consumers are more sensitive to a price change in soy milk compared to that of other plant milk and conventional milk products. All the milk alternative beverages under consideration showed inelastic own-price elasticity demands. Among the significant compensated own price elasticities, 2% milk had the most inelastic elasticity of demand, which was −0.17, meaning that price changes have a relatively small impact on product consumption and it is taken by consumers as the most necessary milk product in their daily lives. Irz and Kuosmanen (2013) [65] estimated a complete system of demand for food and dairy products in Finland and found that all food groups are price inelastic, with the lowest levels of price sensitivity belonging to meat, fish and dairy products. Bouamra-Mechemache et al. (2008) [66] in their review article also demonstrated the inelastic nature of demand for dairy products within the EU, with an average own-price elasticity reported to be −0.57.

17 out of 42 (40 percent) compensated cross-price elasticities had negative signs denoting net complements. There was no significant substitute for almond milk. Soy milk’s net substitutes were all four types of regular milk products and it is a strong net complement for almond milk and rice milk. One possible explanation is that because the data used is purchase data, it means that almond milk and soy milk are normally purchased together. Moreover, the results showed that almond milk and 2% milk are substitutes. Even though the results did not indicate that rice milk could be a substitute for conventional milk, it serves as complements for soy milk and almond milk. As expected, all four types of conventional milk products were strong substitutes between each other. For most product types, the absolute values of the cross-price elasticity were very small, i.e., close to 0, exhibiting that the consumption of most milk products was independent of the price of other similar ones.

Comparatively, in terms of log hedonic metric augmented Barten’s synthetic model, soy milk was also found to be the most expenditure-elastic plant-based milk alternative beverage. There was no strong substitute for almond milk and rice milk. Soy milk and other types of conventional milk products were found to be substitutes for 2% low-fat milk, which is the same as the results we summarized above. Also, soy milk was a substitute for fat-free milk and whole milk. Soy milk and rice milk acted as net complements to almond milk. Not surprisingly, soy milk was continuously a substitute for all conventional milk types. Almond and soy milk and four conventional milk products were all complements of rice milk. The same strong substitution effects could be found among all four conventional milk products.

## 6. Conclusions and Implications

Consumer demand estimation results in this study solidified their substitution effect, offering valid explanations of the trend of shifting consumption from conventional milk to plant milk. This work evaluated the consumer demand for conventional milk and plant milk using cutting-edge modifications to demand system models. A novel product characteristics approach was exploited by introducing qualitative factors through hedonic metric approach in approximating the elasticities estimated by Barten’s synthetic model. Both linear and semi-log hedonic pricing models showed good fitness with the data of seven products. The estimated parameters from Barten’s synthetic model were greatly reduced and significant. Furthermore, the homogeneity and symmetry test results were observed as expected. Soy milk had the highest own-price elasticity. Inelastic demand of all three types of plant milk means that consumers’ purchasing is not very sensitive to price changes. Plus, soy milk was found to be a substitute for all four types of conventional milk products and vice versa. This provides a good explanation for the consumption trend toward plant-based beverages. Additionally, three plant milk were complements among each other.

The estimated elasticities can be used to project future demand trends of plant milk, which is increasingly capturing the attention of startups and food companies to take initiative steps to develop novel products with better quality, attractive packaging and a positive influence on agricultural food market growth. The findings are hoped to shed some light for policy makers to improve the agriculture sector to increase raw material, such as soy, almond and rice, availability. The plant milk market is a classic example of a monopolistically competitive market, so the government should guide the formation of a benign competitive market, prevent a monopoly, and ensure that the price is stable at a reasonable level. Future research can be induced by exploring the competitive behavior of member firms in the market and their performance outcomes including prices, profits, output, etc. Additionally, with the substitution effects between plant milk and conventional milk observed, the conventional milk market requires continuing product alternation and differentiation to improve its competitiveness and sustainability.

Due to the limited availability of nutritional information and recorded consumption data about plant milk, this study could only apply pooled UPC level information to estimate the hedonic pricing models. As plant milk started to gain popularity in the recent ten years, adequate purchase observations were not available at the beginning of the time period in this work. The dataset for estimating hedonic pricing model requires sufficient variations on the nutritional attributes, but the existing household level data cannot satisfy such a variability requirement. Furthermore, the information of nutritional data about plant milk is limited and very rare in the Nielsen Homescan database as well as the USDA nutritional database. The estimated results of demand elasticities could possibly be much more informative if more consumer purchase data and nutritional information about plant milk are available.

## Figures and Tables

**Figure 1 foods-10-00265-f001:**
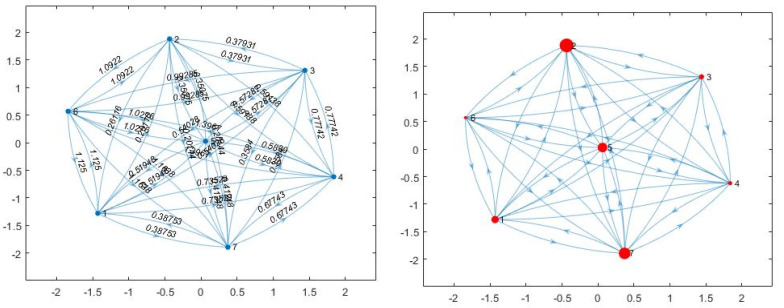
Closeness-index based on linear hedonic distance of seven products.

**Figure 2 foods-10-00265-f002:**
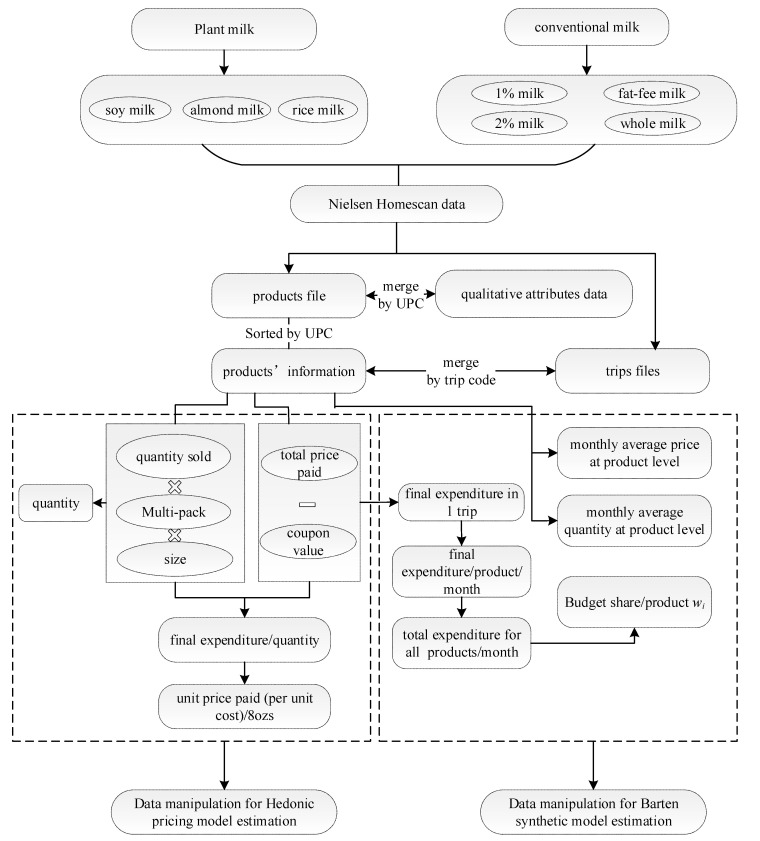
Data manipulation procedure for hedonic pricing models and Barten’s synthetic demand system.

**Table 1 foods-10-00265-t001:** Linear and semi-log hedonic distance matrix.

	Almond Milk		Soy Milk		Rice Milk		2% Milk		1% Milk		Fat-Free Milk		Whole Milk	
	Linear	Log	Linear	Log	Linear	Log	Linear	Log	Linear	Log	Linear	Log	Linear	Log
Almond milk	0.00	0.00	0.26	0.26	0.54	0.51	0.74	0.62	0.52	0.92	0.39	1.30	1.13	0.79
Soy milk	0.26	0.25	0.00	0.00	0.38	0.34	0.59	0.48	0.35	0.97	0.20	1.11	1.09	0.71
Rice milk	0.54	0.51	0.38	0.34	0.00	0.00	0.78	0.64	0.57	1.09	0.36	1.03	0.99	0.73
2% milk	0.74	0.62	0.59	0.48	0.78	0.64	0.00	0.00	0.58	1.11	0.68	1.14	1.40	0.95
1% milk	0.52	0.92	0.35	0.97	0.57	1.09	0.58	1.11	0.00	0.00	0.41	1.66	1.03	0.78
Fat-free milk	0.39	1.30	0.20	1.11	0.36	1.03	0.68	1.14	0.41	1.66	0.00	0.00	1.16	1.42
Whole milk	1.13	0.79	1.09	0.71	0.99	0.73	1.40	0.95	1.03	0.78	1.16	1.42	0.00	0.00

**Table 2 foods-10-00265-t002:** Continuous linear and log hedonic distance matrix.

	Almond Milk		Soy Milk		Rice Milk		2% Milk		1% Milk		Fat-Free Milk		Whole Milk	
	Linear	Log	Linear	Log	Linear	Log	Linear	Log	Linear	Log	Linear	Log	Linear	Log
Almond milk	1.00	1.00	0.79	0.80	0.65	0.66	0.58	0.62	0.66	0.52	0.72	0.43	0.47	0.56
Soy milk	0.79	0.80	1.00	1.00	0.73	0.75	0.63	0.68	0.74	0.51	0.83	0.47	0.48	0.58
Rice milk	0.65	0.66	0.73	0.75	1.00	1.00	0.56	0.61	0.64	0.48	0.74	0.49	0.50	0.58
2% milk	0.58	0.62	0.63	0.68	0.56	0.61	1.00	1.00	0.63	0.47	0.60	0.47	0.42	0.51
1% milk	0.66	0.52	0.74	0.51	0.64	0.48	0.63	0.47	1.00	1.00	0.71	0.38	0.49	0.56
Fat-free milk	0.72	0.43	0.83	0.47	0.74	0.49	0.60	0.47	0.71	0.38	1.00	1.00	0.46	0.41
Whole milk	0.47	0.56	0.48	0.58	0.50	0.58	0.42	0.51	0.49	0.56	0.46	0.41	1.00	1.00

**Table 3 foods-10-00265-t003:** Nearest neighbor dummy matrix ($d_{ij}^{nn})$ of linear and log hedonic distance matrix.

	Almond Milk		Soy Milk		Rice Milk		2% Milk		1% Milk		Fat-Free Milk		Whole Milk	
	Linear	Log	Linear	Log	Linear	Log	Linear	Log	Linear	Log	Linear	Log	Linear	Log
Almond milk	0	0	1	1	0	0	0	0	0	0	0	0	0	0
Soy milk	0	1	0	0	0	0	0	0	0	0	1	0	0	0
Rice milk	0	0	0	1	0	0	0	0	0	0	1	0	0	0
2% milk	0	0	0	1	0	0	0	0	1	0	0	0	0	0
1% milk	0	0	1	0	0	0	0	0	0	0	0	0	0	1
Fat-free milk	0	0	1	0	0	1	0	0	0	0	0	0	0	0
Whole milk	0	0	0	1	1	0	0	0	0	0	0	0	0	0

**Table 4 foods-10-00265-t004:** Joint hypothesis tests for seasonal dummies, lambda and mu in demand system.

Tests	Model	Estimate	*p*-Value	Test Results
Test0	linear	10.56	0.01	d11 = d12 = d13
	log	6.26	0.10	d11 = d12 = d13
Test1	linear	9.19	0.03	d21 = d22 = d23
	log	8.62	0.03	d21 = d22 = d23
Test2	linear	5.87	0.12	d31 = d32 = d33
	log	6.55	0.09	d31 = d32 = d33
Test3	linear	3.91	0.27	d41 = d42 = d43
	log	2.08	0.56	d41 = d42 = d43
Test4	linear	16.88	0.00	d51 = d52 = d53
	log	16.05	0.00	d51 = d52 = d53
Test5	linear	0.87	0.83	d61 = d62 = d63
	log	1.41	0.70	d61 = d62 = d63
Test6	linear	81.86	<0.0001	lambda = 0, mu = 0
	log	99.93	<0.0001	lambda = 0, mu = 0
Test7	linear	1787.5	<0.0001	lambda = 1, mu = 1
	log	1810.1	<0.0001	lambda = 1, mu = 1
Test8	linear	67.41	<0.0001	lambda = 1, mu = 0
	log	79.59	<0.0001	lambda = 1, mu = 0
Test9	linear	1832.4	<0.0001	lambda = 0, mu = 1
	log	1840.7	<0.0001	lambda = 0, mu = 1

**Table 5 foods-10-00265-t005:** Expenditure elasticities and uncompensated own-price and cross-price demand elasticities estimated by linear hedonic Barten’s synthetic model.

	Almond Milk	Soy Milk	Rice Milk	2% Milk	1% Milk	Fat-Free Milk	Whole Milk	Expenditure
almond milk	−0.13 *	−0.06	−0.06 ***	−1.33 ***	−0.61 ***	−0.92 ***	−0.68 ***	3.60 ***
	(0.06)	(0.04)	(0.01)	(0.41)	(0.18)	(0.27)	(0.20)	(0.00)
soy milk	−0.03 ***	−0.50 ***	−0.02 ***	−3.71 ***	−1.62 ***	−2.46 ***	−1.85 ***	10.07 ***
	(0.00)	(0.04)	(0.00)	(0.45)	(0.20)	(0.30)	(0.22)	(1.21)
rice milk	−0.13 ***	−0.19 ***	−0.10	−0.91	−0.47	−0.50	−0.49	2.31
	(0.03)	(0.06)	(0.13)	(0.93)	(0.40)	(0.62)	(0.46)	(2.50)
2% milk	−0.00 ***	−0.02 ***	−0.00 ***	−0.42 ***	−0.11 ***	−0.17 ***	−0.12 ***	0.83 ***
	(0.00)	(0.00)	(0.00)	(0.03)	(0.01)	(0.04)	(0.01)	(0.07)
1% milk	−0.00 ***	−0.03 ***	−0.00 ***	−0.36 ***	−0.33 ***	−0.24 ***	−0.18 ***	1.14 ***
	(0.00)	(0.00)	(0.00)	(0.03)	(0.02)	(0.02)	(0.02)	(0.08)
fat-free milk	−0.00 ***	−0.01 ***	−0.00	−0.15 ***	−0.07 ***	−0.28 **	−0.07 **	0.57 ***
	(0.00)	(0.00)	(0.00)	(0.03)	(0.01)	(0.02)	(0.01)	(0.07)
whole milk	−0.00 ***	−0.01 ***	−0.00 ***	−0.14 ***	−0.06 ***	−0.10 ***	−0.22 ***	0.55 ***
	(0.00)	(0.00)	(0.00)	(0.04)	(0.02)	(0.02)	(0.03)	(0.10)

**Note**: *p*-value = 0.05 for rejecting the null hypothesis; ***, ** and * indicate significance at 0.001, 0.01, 0.05 levels. The value under each estimate is standard error.

**Table 6 foods-10-00265-t006:** Compensated own-price and cross-price demand elasticities estimated by linear hedonic Barten’s synthetic model.

	Almond Milk	Soy Milk	Rice Milk	2% Milk	1% Milk	Fat-free Milk	Whole Milk
almond milk	−0.12 *	−0.13 ***	−0.06 ***	0.01	−0.03 *	−0.02	−0.01
	(0.06)	(0.03)	(0.01)	(0.01)	(0.01)	(0.01)	(0.01)
soy milk	0.00	−0.25 ***	−0.01 ***	0.06 ***	0.02 ***	0.04 ***	0.03 ***
	0.00	0.03	0.00	0.01	0.00	0.01	0.00
rice milk	−0.12 ***	−0.13 ***	−0.01	−0.04	−0.09 ***	0.08	−0.06 ***
	(0.03)	(0.03)	(0.13)	(0.03)	(0.03)	(0.07)	(0.02)
2% milk	0.03	0.00 ***	−0.00	−0.11 ***	0.03 ***	0.04 ***	0.03 ***
	(0.00)	(0.00)	(0.00)	(0.01)	(0.00)	(0.00)	(0.00)
1% milk	−0.00 **	0.00 ***	−0.00 ***	0.06 ***	−0.15 ***	0.04 ***	0.03 ***
	(0.00)	(0.00)	(0.00)	(0.00)	(0.02)	(0.00)	(0.00)
fat-free milk	−0.00	0.00 ***	0.00	0.06 ***	0.03 ***	−0.14 ***	0.03 ***
	(0.00)	(0.00)	(0.00)	(0.00)	(0.00)	(0.01)	(0.00)
whole milk	−0.00	0.00 ***	−0.00 ***	0.06 ***	0.03 ***	0.04 ***	−0.12 ***
	(0.00)	(0.00)	(0.00)	(0.01)	(0.00)	(0.00)	(0.02)

**Note:***p*-value = 0.05 for rejecting the null hypothesis; ***, ** and * indicate significance at 0.001, 0.01, 0.05 levels. The value under each estimate is standard error.

## Data Availability

Restrictions apply to the availability of these data. Part of the Data was obtained from Nielsen Company (U.S.), LLC and Kilts Center for Marketing Data Center at the University of Chicago Booth School of Business] and are available from the authors with the permission of Nielsen Company (U.S.), LLC and Kilts Center for Marketing Data Center at the University of Chicago Booth School of Business.

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
