# Peer review of "U.S. Consumer Demand for Plant-Based Milk Alternative Beverages: Hedonic Metric Augmented Barten’s Synthetic Model"

_foods, 2021, doi:10.3390/foods10020265_

Round 1

Reviewer 1 Report

The paper deals with a relevant issue concerning preferences for alternative foods with different contributions to the diet.

The mathematical approach is extensively described, while the core of the results is very limited (lines 519-547, page 9). In my opinion, the mathematical description could be reported in supplemental material and the core of the paper could be extended also discussing the nutritional components of the hedonic space referring to the previous Yang and Dharmasena 2020 paper.

Author Response

Thanks for giving me the valuable advice. According to your suggestions, I have revised the empirical results and discussion part (section 5) to include more discussion related to the results of hedonic pricing model referring to the previous paper (revision can be found from line 389 to line 415 in the revised manuscript). Also, I put the derivation of the expression of demand elasticities and some figures describing the method to the appendices to leave more room for the results discussion.

Reviewer 2 Report

Author of the manuscript entitled "U.S. Consumer Demand for Plant-Based Milk Alternative Beverages: Hedonic Metric Augmented Barten Synthetic Model" presented the review work appropriately. However, there are some minor errors in the text which need to address prior to its publication in the journal.

Line 57: "As shown in 2016?.... Shown by whom? write the complete citation.

Line 543: "To conclude"....Author already mentioned it in the Section 6. "Conclusion and Implications" . Therefore, it is not necessary to write double conclusion.

Section 6: "Conclusion and implication" is too long.

Rewrite the results and discussion citing more recent references from the similar field of study. 

Author Response

Point 1: Line 57: "As shown in 2016?.... Shown by whom? write the complete citation.

Response 1: The complete citation has been given in the revised manuscript in line 57: Copeland and Dharmasena (2016)

Point 2: Line 543: "To conclude"....Author already mentioned it in the Section 6. "Conclusion and Implications" . Therefore, it is not necessary to write double conclusion.

Response 2: Thanks for giving me this advice. I have deleted the repeated conclusions in section 5.

Point 3: Section 6: "Conclusion and implication" is too long.

Response 3: Thanks so much for your valuable advice. I have revised the conclusion and implication part to make it more concise. According to another reviewer’s suggestion to add more information explaining why this paper can inspire future research or implications for practice, I have compromised the two suggestions to make this part as much concise as I can.

Point 4: Rewrite the results and discussion citing more recent references from the similar field of study

Response 4: Thanks for giving me this suggestion. I have rewritten the results and discussion part to put it into two sub-sections and added more discussion about the results of hedonic pricing model (form line 386 to line 419). Also, according to your suggestion, I cited many recent references from similar research field which can be seen in the revised manuscript.   

Reviewer 3 Report

Thanks to the author(s) for a well-researched manuscript. The paper has potential if a revision can address a few major and minor suggestions.

The major suggestions relate to the methodology and justification. The paper needs a stronger argument throughout and why the study is important. In the introduction section, the author(s) should provide a lot stronger argument.

The framing of the work needs a broader conceptual motivation. For instance, the literature review was scattered and mostly offered a list of precedents in the literature to justify the choices of materials/variables. Instead, in your introduction, focus on building a conceptual

framework and make it clear to the reader what theoretical points are at stake in your research.

A lot detailed information/explanation should be added into “Data” section.

Conclusion: The discussion is a little disappointing and could be written in a much stronger way. Also, there are very little managerial implications, and more specific, poignant recommendations should be provided to the practitioners/managers. This information would give the paper a much better finish.

There is no answer to the "so what" question

There is nothing here to inspire future research or implications for practice

The results are clear, but nothing new. The findings do not provide useful guidelines for scholars/practitioners.

Author Response

Point 1: In the introduction section, the author(s) should provide a lot stronger argument about why the study is important

Response 1: Thanks for your valuable suggestions. I have revised the introduction and provided stronger argument to explain why the study is important from line 107-132 in the revised manuscript. Also, in the conclusions and implications (section 6), I added some argument discussing the importance of the estimation results from this research from line 559 to 570.

Point 2: In your introduction, focus on building a conceptual framework and make it clear to the reader what theoretical points are at stake in your research.

Response 2: Thanks so much for your advice. I have established a conceptual framework introducing the theoretical points in this research from line 91 to 106.

Point 3: A lot detailed information/explanation should be added into “Data” section.

Response 3: Thanks again for your suggestions. I revised the data section and added a lot more detailed information to explain the data used for estimating hedonic pricing model and demand system from line 330 to 362 and from line 364 to 380 in the revised manuscript.

Point 4: There is no answer to the "so what" question and there is nothing here to inspire future research or implications for practice.

Response 4: As responded in point 1, I revised the conclusion and implication section attempting to answer the “so what” question and made the implications of the results much stronger to show why this research can inspire future research and why it is meaningful for practitioners/managers and policy makers (from 559 to 570). Considering another reviewers’ advice to shorten the conclusion and implication section, I compromised these two opinions to try to make this part more concise and clearer.

Round 2

Reviewer 2 Report

The authors of the manuscript have revised and corrected the "foods-1052619"

carefully and responded to all the comments appropriately.

Now, the manuscript can be considered for publication.